# Factors associated with the timely completion of doctoral research studies in clinical pharmacy: A mixed-methods study

**Adaobi U. Mosanya**[1], **Blessing O. Ukoha-kalu**[1,2]*, **Abdulmuminu Isah**[1], **Ifeoma Umeh**[3], **Kosisochi C. Amorha**[1], **Ebere E. Ayogu**[1], **Chukwuemeka Ubaka**[1]

**1** Department of Clinical Pharmacy and Pharmacy Management, Faculty of Pharmaceutical Sciences, University of Nigeria Nsukka, Nsukka, Enugu State, Nigeria, **2** Hull York Medical School, University of Hull, Hull, England, United Kingdom, **3** Nnamdi Azikiwe University, Awka, Anambra State, Nigeria

* b.ukoha-kalu@hull.ac.uk

**Data Availability Statement:** All relevant data are within the article and its Supporting information files.

## Abstract

### Introduction

There is growing scientific evidence of mental and well-being issues that doctoral research students face as a result of not finishing their program on time. This study aims to explore the factors associated with the timely completion of doctoral research studies in the clinical pharmacy speciality.

### Methods

This was a mixed-method study that combined surveys with in-depth interviews. Current doctoral research students and pharmacists who have recently completed their doctoral research program participated in the study. A validated questionnaire and an interview topic guide developed from the literature and pretesting were used to collect data. Data for this study were collected between February 2021 and September 2021. Quantitative data were analysed with the Statistical Package for Social Sciences (SPSS) V.25 while interview data were subjected to reflexive thematic analysis.

### Results

47 students who are currently pursuing their doctoral research program in clinical pharmacy participated in the survey, while 8 pharmacists who had recently completed their doctoral research program in clinical pharmacy participated in the in-depth interviews. Five themes were identified: factors contributing to delay in the program, factors contributing to the timely completion of the program, ways to improve the program, advice to current students and advice to prospective students. Having more than one supervisor, supervisors' commitment to the research work and support from the department were identified as facilitators of timely completion of doctoral research programs in clinical pharmacy.

**Funding:** The author(s) received no specific funding for this work.

**Competing interests:** The authors have declared that no competing interests exist.

## Conclusion

Our study provides an understanding of the barriers and facilitators of timely completion of doctoral research programs in the clinical pharmacy specialist, and how these can be used to improve the postgraduate study programs in Nigeria.

## Introduction

Societies are presented with complex issues that necessitate both immediate and long-term solutions. Through research, doctoral students have a major role to play in providing these solutions, as they are the new generators of knowledge. Over the years, doctoral studies both in developed and developing countries have shown an identical trend; a smaller percentage of PhD students graduate within the due date. In many African universities, a doctoral research programme typically takes about 6 years to complete [1]. In Nigeria, full-time studies take about three years and part-time studies take four years, with less than 10% of students completing their studies on time [2].

Availability of time, supervisor's commitment to the project, and family commitments have been attributed to be the constraints to the timely completion of postgraduate studies in Nigeria [3]. Timely completion in this context refers to the submission of the thesis for assessment within the time specified by the program. There is a need to examine timely doctoral studies because of the increase in the number of doctoral students, the costs and consequences of not completing doctoral studies on time, and a growing concern about doctoral students' well-being [4, 5]. Untimely completion of doctoral research studies can lead to a loss in income, productivity, and knowledge base for the educational institution [6]. For students, failing to complete their Doctorate research programmes on time could result in job loss or promotion delays, as well as regret, frustration, and disappointment [2]. Timely completion of a postgraduate program is linked to the university's reputation and this can affect the choice of the university by prospective students [7]. There is growing scientific evidence of mental and well-being issues that doctoral research students face as a result of not finishing their program on time [8, 9].

It is believed that the department micro-culture is a contributing factor to the timely completion of doctoral research programs in Nigeria [10]. The department has a significant impact on the timely completion of doctoral degree programs because it is a decentralised body within the institution with its admission procedures [11]. The supervisors' commitment to the research project and the relationship between the supervisor and student are believed to also influence the timely completion of the program [11]. At the time of this study, no study had examined the factors that could affect the timely completion of a doctoral research program in a clinical pharmacy speciality. This study aimed to assess factors that influence the timely completion of doctoral studies in the clinical pharmacy specialization and evaluate the association of these factors with supervisor and departmental support.

## Methods

### Study design and participants

This was a mixed-method study using, i) a survey, and ii) semi-structured interviews with current and past doctoral research students in clinical pharmacy. Phase I was a cross-sectional study among clinical pharmacy doctoral research students in eight accredited schools of

pharmacy in Nigeria, while Phase II was an in-depth interview of clinical pharmacy doctoral students who had just completed their program.

## Setting

Phase one was conducted among current doctoral research students in accredited pharmacy schools in the south-east, south-south, south-west, north-central and north-west Nigeria. Phase II was conducted among doctoral students of the University of Nigeria, who recently completed their doctoral research program.

## Eligibility criteria

In Phase I, participants were current doctoral research students from accredited pharmacy schools in Nigeria, while in Phase II, participants were immediate past doctoral students from the University of Nigeria Nsukka, a public university in southeast Nigeria.

## Sampling strategy

The survey data were collected for convenience, but the in-depth interview participants were invited using a purposive sampling framework based on those who recently completed their doctoral research program.

## Participant recruitment and data collection

In Phase I, the survey was completed through an online questionnaire. In phase II, the interviews were conducted face-to-face by a member of the research team. Participants were informed about the study during pre-assessment. If they were interested and accepted participation, they completed a consent form with or without the assistance of a member of the study team, according to their preference. Phase 1: a validated questionnaire which was modified to suit the Nigerian context [12], was adopted for this study. The questionnaire consists of 3 domains: research progress status, lead supervisor support, and department support. Online survey data was collected between March 2021 –April 2021. Phase II: An interview topic guide was developed from the literature and through pre-testing. Contributory factors to delay in program completion, contributory factors to timely completion of doctoral research, and possible ways to improve the program were explored. Interviews were audio-recorded and then transcribed verbatim and anonymised. Interviews for Phase II were conducted between July 2021 –September 2021.

## Data analysis

The survey was analysed, using descriptive statistics, in Statistical Program for Social Sciences (SPSS) v.25. Interview transcripts were analysed using thematic analysis [13], with six steps: familiarization, coding, searching for themes, reviewing themes, defining themes, and reporting [13]. Thematic analysis was chosen because it is flexible, can summarize significant elements of a big data set while also providing a "thorough description," and allows for both social and psychological interpretation [13]. NVivo (V.12) software was used to manage the interview data.

## Ethical considerations

This study was reviewed and approved by the Faculty of Pharmaceutical Sciences ethics review board, University of Nigeria Nsukka. The aim of the research and interview method was explained to the participants. All the participants received written information about the

project and gave both verbal and written consent to participate according to the procedures of research ethics clearance. All personal identifiers were removed during analysis, and responses from the participants were treated with confidentiality.

## Results

### Demographic characteristics of participants

Phase I: 47 participants completed the online survey questionnaire. Phase II: Eight immediate past doctoral students participated in the in-depth interviews. Table 1 shows the demographic characteristics of the participants in Phase I. The mean age of the respondents was 39.5 ± 4.3 years. More than one-third of the participants were from south-east Nigeria.

### Research progress status, lead supervisor support and department support

In Phase I, participants had low research status, lead supervisor support and department support scores. Details of research status, lead supervisor support and department support scores are shown in Table 2.

**Table 1. Sociodemographic variables of the respondents (N = 47).**

|  | Mean | Standard deviation |
|---|---|---|
| **Age** | 39.5 | 4.3 |
|  | **Frequency** | **Percentage (%)** |
| **Gender** |  |  |
| Male | 24 | 51.1 |
| Female | 23 | 48.9 |
| **Marital status** |  |  |
| Single | 7 | 15.0 |
| Married | 38 | 80.8 |
| Widowed | 1 | 2.1 |
| Divorced | 1 | 2.1 |
| **The geographical location of the institution** |  |  |
| South-east | 19 | 40.4 |
| South-south | 12 | 25.5 |
| South-west | 7 | 15.0 |
| North-central | 5 | 10.6 |
| North-west | 4 | 8.5 |
| **Nature of program** |  |  |
| Part-time | 23 | 48.9 |
| Full time | 24 | 51.1 |
| **The staff of the department** |  |  |
| Yes | 21 | 44.6 |
| No | 26 | 55.4 |
| **Number of PhD Supervisors** |  |  |
| One | 11 | 23.4 |
| Two | 32 | 68.1 |
| More than two | 4 | 8.5 |
| **Previously obtained a degree from the same university** |  |  |
| Yes | 22 | 46.8 |
| No | 25 | 53.2 |

**Table 2. Research progress status, lead supervisor support and department support.**

|  | Mean ± SD | Median (IQR) | Range |
|---|---|---|---|
| **Research Status score** | 38.1 ± 9.5 | 39 (14) | 16–55 |
| **Lead Supervisor support score** | 27.7 ± 6.4 | 29 (7) | 10–40 |
| **Department support score** | 23.4 ± 5.5 | 24 (7) | 7–34 |

IQR: Interquartile range

SD: Standard deviation

## Association between the independent variables, the research status, lead supervisor support and department support

Participants who had more than one supervisor reported higher research output. In addition, participants who had obtained a previous degree from the same university reported higher lead supervisor support and department support. Research status scores were also observed to be been positively and highly correlated with lead supervisor and departmental support scores (r = 0.83, p<0.001; r = 0.87, p<0.001). Details of this result are shown in Table 3.

### Study themes

From the analyses of the interviews, five themes were generated: contributory factors to delayed program completion, contributory factors to timely program completion, ways to improve the doctoral research program, advice to current students and advice to prospective students. Table 4 shows more detail on subthemes.

**Theme 1:** Contributory factors to delayed programme completion

In the interview, participants reported that delays in research topic selection and approval contributed to delays in completing the program.

*". . .I think what delayed me was getting a research topic. . ."*

[Participant 01]

*". . .getting a topic, being sure of your research is the major reason because coming in for the PhD, I didn't know what I was coming in to do. I wasn't sure of a topic and then when I*

**Table 3. Associations between the independent variables, the research status, lead supervisor support and department support.**

|  | High Research status N = 25 | Good Lead Supervisor support N = 27 | Good Department support N = 25 | r |
|---|---|---|---|---|
|  | n (%) | n (%) | n (%) |  |
| **Number of PhD Supervisors** |  |  |  |  |
| One | 9(36)* | 10(37)* | 10(40) | 0.83 |
| Two | 13(52) | 14(52) | 11(44)** |  |
| More than two | 3(12) | 3(11) | 4(16) |  |
| **Obtained previous degree(s) from the same university** |  |  |  |  |
| Yes | 11(44) | 9(33)** | 9(36)* | 0.87 |
| No | 14(56) | 18(67) | 16(64) |  |

[a]Chi-square

*significant at p < 0.05

**significant at p<0.001

**Table 4. A list of Themes and subthemes.**

| Themes | Subthemes |
|---|---|
| Contributory factors to delayed program completion | Delay in topic selection and approval |
|  | Lack of sufficient study leave |
|  | Financial difficulties |
|  | Strike and COVID-19 pandemic lockdown |
| Contributory factors to timely program completion | The rapport between the student and supervisor |
|  | Supervisors' trust in the student |
|  | Support from department |
|  | Strike and COVID-19 lockdown |
| Ways to improve the doctoral research program | Time management by the department |
|  | Bridging the gap between pharmacists in the academia and the hospital |
|  | Course match according to the area of interest |
| Advice to current students | Seek support from experts<br>Work, life and study balance |
| Advice to prospective students | Getting research topic before enrollment |
|  | Financial readiness |
|  | Readiness for innovation and hard work |

*eventually got one, the feasibility study just showed me that I had to forfeit that. And changed the topic. So that took like a year."*

[Participant 02]

Not being on study leave also affected the timely completion of the program. Two participants reported that they were not granted study leave by their place of work. One of the participants working in another institution as a lecturer stated that he was not granted study leave by his place of work.

*"I was supposed to be on full-time study leave but I was not granted full-time by my place of work. I am still carrying my full work and still running my PhD at the same time."*

[Participant 03]

*"I got study leave for the 3 years but we have to stay fully at work because we were the only ones teaching at the department . . ."*

[Participant 04]

One of the participants working in the same department reported how difficult it is to get a study leave from the university administration.

*"It is difficult collecting study leave as a research student. . . People are so afraid to apply for study leave because when you apply they say, it is without pay."*

[Participant 05]

Insufficient funding also delayed the program according to some of the respondents:

*"It was draining financially. . ."*

[Participant 05]

*"I would say the cost of the study. . . everything is costly. Only laboratory investigation took up to all my savings"*

[Participant 06]

The COVID-19 pandemic, and strike action by the Academic Staff Union of Universities (ASUU) also contributed to the delay in the program:

*"One year was just lost due to COVID-19. I can say It's now 4 years; an extra one year instead of 3 years."*

[Participant 04]

*"The COVID-19 pandemic delayed me."*

[Participant 07]

*"Then strike added to the delay. . ."*

[Participant 08]

**Theme 2:** Contributory factors to timely programme completion

Two of the participants completed the program within the time frame required. They agreed that some of the contributory factors were: the rapport they had with their supervisors, supervisors' trust in their decisions and support from the department.

*"Well the major factor that helped me was the rapport I had with my supervisors. . ."*

[Participant 05]

*"My two supervisors were there, they trusted me so much. I designed my work, collected data, and I d my analysis from the beginning to the end. They gave me that trust. I believe that was part of what helped me to also complete on time."*

[Participant 05]

Surprisingly, the COVID-19 pandemic also contributed to the timely completion of the program:

*". . ..as an academic and you're running a PhD programme in the same institution. It is so difficult when you want to tell your HOD that you want to travel for 6 months to collect data. Anybody will roll their eyes for you. If not for the COVID-19 pandemic, I wouldn't have finished my work by now. It would have been difficult for me to collect data."*

[Participant 06]

*". . .But if not for the COVID-19 pandemic, I don't think my institution would have allowed me to travel for data collection from Feb to November. . ."*

[Participant 05]

**Theme 3:** Ways to improve the PhD programme

Some of the participants believed that time management and support from the departments can help students to complete their doctoral research within the time frame:

*"Everybody tries to ensure that the undergraduates finish their program on record time. But the post-graduate program is like you have to be the one to push."*

[Participant 02]

*"There is nothing that stagnates a research program more than the supervisor being too busy. If the supervisor is so committed that he hardly looks at your work that will create a lot of problems. As my work has been ready for the past 6 months but my supervisor did not have time to look at it."*

[Participant 01]

Some of the participants suggested that coursework for doctoral studies should be based on the students' area of interest:

*". . .if a student is specialising in Pharmacoeconomics, he should be taught courses in Pharmacoeconomics. . .. courses should be tailored specifically towards the students' research area"*

[Participant 04]

**Theme 4:** Advice to current students

Some of the participants agreed that "working with timelines" helped them to complete their research:

*". . .Timeline is important. . . Make sure you do something every day. . . Because before you know it you have completed the research. . .."*

[Participant 01]

*". . .Write down your goals. . . And put a time frame to it. . ."*

[Participant 07]

Seeking support from experts in your field also helped some of the participants to complete their project research:

*". . .talk to people, observe, because you learn a lot by observing the things around you. . . then be open to suggestions, criticism, constructive ones I mean. With that, you won't have any problem."*

[Participant 04]

Consequently, having a good work, life and study balance is vital to the timely completion of a doctoral research program:

"balancing the challenge of work or family life with that of academic is very important. . ."

[Participant 03]

**Theme 5:** Advice to prospective doctoral research students

Getting a research topic before enrolling in the program can help facilitate data collection and, ultimately, timely completion of the program:

*". . .even before applying for doctoral research, you should have an idea of what you want to do. That is the first thing. . . Not having an idea of what to do can cause unnecessary delays."*

[Participant 08]

*". . .get a research topic before you commence the programme. it will make things faster. . ."*

[Participant 02]

According to some of the participants, saving up for the research can help facilitate the process:

*". . . If you are not financially ready, it will stagnate the work. . ."*

[Participant 01]

Some of the participants agreed that "*readiness to innovate*" and "hard work" are essential for the timely completion of any doctoral research program:

*". . .It is a program that requires hard work. . ."*

[Participant 03]

*"..Be ready to task your brain and be innovative. . ."*

[Participant 05]

## Discussion

This qualitative study aimed to explore the factors that can contribute to the timely contribution of a doctoral research program in a clinical pharmacy specialty. The majority of the participants agreed that delays in topic selection by the student, topic approval by the department, not being granted study leave, and financial constraints could contribute to delays in the timely completion of the program. Support from the department, the rapport between the student and supervisor, and the supervisors' commitment to the research work were identified as factors that can help doctoral students to complete their program on time. Prospective students were advised to settle for a research topic even before enrolling on the program. This study provides valuable insight into what could be done to minimize delays in the timely completion of a doctoral research program in the clinical pharmacy specialty.

First, having more than one supervisor increased the research status of the participants. A recent study has reported a positive correlation between having more than one supervisor and a reduced rate of intention to quit the program [14]. These are consistent with our findings–students who had more than one supervisor experienced a reduced risk of experiencing burnout. Second, students who received support from their supervisors and department were able to complete their program within the required time. Our findings are consistent with previous work acknowledging the role of the department and supervisor in the timely completion of any postgraduate program [15]. Third, exposing the students to topics that are related to their

research interests will help the students navigate easily during the different stages of the research. It is essential to consider the needs of the students and ensure that each student gets the support they need to complete their research [16].

Some areas of refinement were identified in this study. Participants explained the need for timely approval of the research topic by the department. Approval of a research topic is believed to be the first step in carrying out any research. There is a need to look for ways to improve supervisors' commitment to the research work. Supervisors' commitment to the research project is a major factor that can motivate the student to work hard while influencing the timely completion of the program [17, 18]. The split between pharmacists in academia and hospital pharmacists, the divide between students' areas of interest and supervisors' areas of expertise, and insufficient communication between postgraduate students and departments are all barriers to the timely completion of doctoral research in the clinical pharmacy speciality [2, 18–20].

## Strength and limitations of the study

Our study contributes to the growing body of evidence that supports how timely completion of postgraduate studies may benefit the students, universities, and the society [2, 7], with emphasis on the supervisor's commitment, support from the department, and approval of study leave by the institution.

The qualitative data acquired through in-depth interviews represent the narratives of a sample of participants and therefore may not be more widely representative. In addition, the postgraduate program offered by these universities in Nigeria may differ in other countries.

## Conclusion

Our study provides an understanding of the barriers and facilitators of timely completion of doctoral research programs in the clinical pharmacy specialty, and how these can be used to improve the postgraduate study programs in Nigeria.

## Supporting information

**S1 File.**
(DOCX)

## Acknowledgments

The authors wish to thank the current and immediate past doctoral students who participated in the study.

## Author Contributions

**Conceptualization:** Adaobi U. Mosanya, Blessing O. Ukoha-kalu, Ifeoma Umeh, Kosisochi C. Amorha, Ebere E. Ayogu.

**Data curation:** Adaobi U. Mosanya, Ifeoma Umeh, Chukwuemeka Ubaka.

**Formal analysis:** Adaobi U. Mosanya, Blessing O. Ukoha-kalu.

**Investigation:** Adaobi U. Mosanya.

**Methodology:** Adaobi U. Mosanya, Ebere E. Ayogu.

**Project administration:** Adaobi U. Mosanya.

**Writing – original draft:** Blessing O. Ukoha-kalu.

**Writing – review & editing:** Adaobi U. Mosanya, Blessing O. Ukoha-kalu, Abdulmuminu Isah, Ifeoma Umeh, Kosisochi C. Amorha, Ebere E. Ayogu, Chukwuemeka Ubaka.

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
