## [Decision Letter · Decision Letter 0]

12 Jul 2022

PONE-D-22-17270Factors associated with the timely completion of doctoral research studies in clinical pharmacy: a mixed methods studyPLOS ONE

Dear Dr. Ukoha-kalu,

Thank you for submitting your manuscript to PLOS ONE. After careful consideration, we feel that it has merit but does not fully meet PLOS ONE’s publication criteria as it currently stands. Therefore, we invite you to submit a revised version of the manuscript that addresses the points raised during the review process.

ACADEMIC EDITOR: 

Well written article.

May be asked for data availability as per the policy of journal.

Authors must address the points raised by  the reviewer #1.

Sample size calculation not done by authors. Authors are requested to do so. 

The author only uses one university as the sample. Authors are requested to please write about the policy of the university. also need comments on variables.

We look forward to receiving your revised manuscript.

Kind regards,

Priti Chaudhary, M.S.

Academic Editor

PLOS ONE

Journal Requirements:

Reviewers' comments:

Reviewer's Responses to Questions

**Comments to the Author**

1. Is the manuscript technically sound, and do the data support the conclusions?

Reviewer #1: Yes

Reviewer #2: Yes

2. Has the statistical analysis been performed appropriately and rigorously? 

Reviewer #1: Yes

Reviewer #2: Yes

3. Have the authors made all data underlying the findings in their manuscript fully available?

Reviewer #1: No

Reviewer #2: Yes

4. Is the manuscript presented in an intelligible fashion and written in standard English?

Reviewer #1: Yes

Reviewer #2: Yes

5. Review Comments to the Author

Reviewer #1: 1. Please relate the outcome of the study with author(s) country policy.

2. Please relate the outcome of the results with the unified theory.

3. The author only uses one public university as the sample. Please justify.

4. Please justify why the author(s) used 47 respondents. Is the sample meet based on your population. Please justify.

5. The literature in the manuscript needs to be improved. Please add more literature about your independent and dependent variables.

Reviewer #2: Well written article. The article is very relevant in different settings across the world.

The introduction is well written. Statistical analysis is detailed and appropriate to the objectives of the study.

The themes and sub-themes have been well brought out.

6. PLOS authors have the option to publish the peer review history of their article (what does this mean?). If published, this will include your full peer review and any attached files.

Reviewer #1: No

Reviewer #2: No

---

## [Author Response · Author response to Decision Letter 0]

16 Aug 2022

1

Comment: Please ensure that your manuscript meets PLOS ONE's style requirements. 

Response: Manuscript revised to meet PLOS ONE’s style requirements.

2.

Comment: In your revised cover letter, please address the following prompts: If there are ethical or legal restrictions on sharing a de-identified data set.

Response: This has been addressed in the revised cover letter

3.

Comment: Please relate the outcome of the study with the author's (s) country policy. 

Response: This has been added to the ‘discussion’

4.

Comment: The author only uses one public university as the sample. Please justify. 

Response: Phase one was conducted among current doctoral research students in accredited pharmacy schools in the south-east, south-south, south-west, north-central and north-west Nigeria. Phase two was conducted among doctoral students of the University of Nigeria, who recently completed their doctoral research program. 

The public university used for phase two is one of the first universities that started postgraduate programs in clinical pharmacy. Also, the University records the highest number of doctoral students in Clinical pharmacy. 

5.

Comment: Please justify why the author(s) used 47 respondents. Is the sample meet based on your population. Please justify. 

Response: 47 respondents were used in phase one. In phase one, study sample = study population

6.

Comment: The literature in the manuscript needs to be improved. Please add more literature about your independent and dependent variables. 

Response This has been revised in the ‘introduction’ and ‘discussion’ sections.

---

## [Editor Report · Decision Letter 1]

1 Sep 2022

Factors associated with the timely completion of doctoral research studies in clinical pharmacy: a mixed methods study

PONE-D-22-17270R1

Dear Dr. Blessing Onyinye Ukoha-kalu

We’re pleased to inform you that your manuscript has been judged scientifically suitable for publication and will be formally accepted for publication once it meets all outstanding technical requirements.

Kind regards,

Priti Chaudhary, M.S.

Academic Editor

PLOS ONE
---

## [Editor Report · Acceptance letter]

22 Sep 2022

PONE-D-22-17270R1 

Factors associated with the timely completion of doctoral research studies in clinical pharmacy: a mixed-methods study 

Dear Dr. Ukoha-kalu:

I'm pleased to inform you that your manuscript has been deemed suitable for publication in PLOS ONE. Congratulations! Your manuscript is now with our production department. 

Kind regards, 

on behalf of

Dr. Priti Chaudhary 

Academic Editor

PLOS ONE